# Study of the Self-Polymerization of Epoxy/Phthalonitrile Copolymers and Their High-Performance Fiber-Reinforced Laminates

**DOI:** 10.3390/polym15173516

**Published:** 2023-08-23

**Authors:** Mingzhen Xu, Bo Li, Xiongyao Li, Zexu Fan, Dengxun Ren

**Affiliations:** 1Yangtze Delta Region Institute (Huzhou), University of Electronic Science and Technology of China, Huzhou 313001, China; 2School of Materials and Energy, University of Electronic Science and Technology of China, Chengdu 610054, China

**Keywords:** amino phthalonitrile, epoxy resin, curing behaviors, fiber-reinforced composites, mechanical properties

## Abstract

Self-polymerization epoxy/phthalonitrile (APPEN) pre-polymers were studied systematically, and then, gelation time and differential scanning calorimetry (DSC) were employed to investigate their curing behaviors. Taking advantage of orthogonal test analysis, the key factors that affected the co-polymerization of APPEN were defined and the appropriate pre-polymerization conditions were analyzed. A possible curing mechanism of APPEN was proposed. Then, the thermomechanical and mechanical properties of glass-fiber-reinforced APPEN laminates (APPEN/GF) prepared at 180 °C were analyzed to understand the cross-linked and aggregation structures. Fracture surface of the composite laminates was also investigated to reveal the copolymerization degree and the interfacial binding. The results indicated that APPEN/GF composites exhibit outstanding mechanical and thermomechanical properties (flexural strength, 712 MPa, flexural modulus, 38 GPa, and *T_g_* > 185 °C). The thermal stability (*T_5%_* > 334 °C and *IPDT* reached 1482 °C) of APPEN/GF composites was also investigated to further reveal the copolymerization between epoxy resin and aminophthalonitrile, which may be beneficial to the application of epoxy-matrix-based composites in the field of high-performance polymer composites.

## 1. Introduction

As an important thermosetting matrix, epoxy resins possess excellent mechanical properties, good adhesion, low cost, and chemical stability, which means they are applied in various fields, including aerospace, mechanics, electronics, ships, and so on [1,2,3]. With the rapid development of aerospace and defense equipment in recent years, the speed of aircraft has sharply increased. Since temperature is positively correlated with speed, higher temperature tolerance is demanded for the structural materials of aircrafts. Limited by their fracture brittleness, poor fire resistance, and low thermal deformation temperature, epoxy resins cannot satisfy the application requirements in high-performance application fields [4,5,6].

To improve the thermal stability of epoxy resins, researchers have carried out many studies on designing curing agents and fillers. Through the optimization of catalysts and fillers, including carbon fiber [7], glass fiber (GF) [8], nanoparticles [9], and organic–inorganic hybrid materials [10], epoxy-based composites exhibit improved thermal stability and mechanical properties. Among them, the introduction of fillers will bring new problems, such as interfacial problems, resulting in the destruction of the structural properties of composite materials. Thus, it is a more economical and efficient strategy to improve the thermal stability of epoxy-resin-based composites by regulating their own polymerization reaction via the introduction of curing agents or copolymer resins. Curing agents, including aromatic amines [11], organic acids [12], phenols [13], and their complexes [14], are usually used and can effectively promote the polymerization of epoxy resins. Aromatic amine curing agents play an important role in the polymerization processes of epoxy resins, due to the obvious improvement in thermal stability for composites, by participating in the main chains of epoxy polymers [14,15].

In our previous work, various phthalonitriles were designed and prepared. Attributable to their high glass transition temperatures (*T_g_*), good dimensional stability, superior flame-resistant properties, and so on [16,17,18], phthalonitriles have aroused extensive interest both from academia and industry. However, the melting temperature and curing temperature of phthalonitriles, which dominate the polymerization processes, are too high and will limit their wide applications. Thus, an amino phthalonitrile monomer was fabricated on the basis of low melt temperature (134 °C). Then, the amino phthalonitrile monomer (aminophenoxyphthalonitrile, APN) was employed as a kind of aromatic curing agent to improve the processability of traditional bisphthalonitrile resins [19]. The results indicated that the polymer composites exhibited excellent thermal and thermal oxidative stability.

Considering the general brittleness of thermosetting resins, to improve the toughness of epoxy resins, various thermoplastic polymers [20,21], nanoparticles [22,23], and other thermosetting resins were introduced and reported [24,25]. For example, Tao et al. [26] reported that poly (methyl methacrylate)-b-poly (butylacrylate)-b-poly (methyl methacrylate) (MAM) can be used to toughen epoxy resin, and the results showed that the impact strength of the composites was increased by 83.5%. Dong et al. [27] reported that CTBN rubber can be used to toughen phenolic epoxy resin, and the glass transition temperature (*T_g_*) of the composite material would decrease with an increase in the content of CTBN. Moreover, controlling the content of CTBN could achieve the best toughening effect when the mass fraction of CTBN was 15%. However, the toughening modification will lead to a further reduction in heat resistance due to the poor heat resistance of toughening modifiers themselves. Herein, a novel toughening effect of rigid group-containing thermoplastic resin Poly (arylene ether nitrile) (PP-PEN) was employed and investigated.

Herein, to expand the various applications of epoxy resin (4,4′diaminodiphenylmethane tetraglycidylamine, AG80), AG80/APN prepolymer toughening with PP-PEN, labeled as APPEN, was studied in detail. The curing behaviors and possible copolymerization mechanism of the as-prepared prepolymers were investigated. Then, glass-fiber-reinforced matrix laminates (APPEN/GF) were fabricated, and the mechanical properties, as well as thermal stability, were systematically studied. The results indicated that the self-promoted copolymerization between epoxy and amino phthalonitrile resins can be adjusted by controlling the content of phthalonitrile, and their composite laminates possess excellent thermal and mechanical properties.

## 2. Experiments

### 2.1. Materials

APN was prepared in our lab. Epoxy resin (4,4′diaminodiphenylmethane tetraglycidylamine, AG80) was kindly supplied by Shanghai Huayi resin Co., Ltd. (Shanghai, China). Poly (arylene ether nitrile) (PP-PEN) was synthesized according to the reference reported previously [28]. Glass fibers were provided by Shenyang Gaote glass fiber Co., Ltd., Shenyang, China. The model was EW170-100. All other reagents were obtained from Chron chemicals (Chengdu, China) and used without further purification.

### 2.2. Preparation of APPEN Prepolymers

According to the orthogonal test design method, APPEN prepolymers were fabricated. Weight of AG80, weight of APN, and prepolymerization time, with three levels for each factor, are shown in Table 1 for nine samples. For Sample 1, AG80 (5 g), APN (4 g), and PP-PEN (1 g) were added to a 100 mL three-necked flask and mixed with DMF as the solvent. Then, the solution was stirred at 140 °C for 20 min to obtain the APPEN prepolymer gel solution. Subsequently, the gel solution was dried at 80 °C and brown solid powder prepolymer obtained (Sample 1, labeled as A5P4T20). The corresponding prepolymers were fabricated using a similar method and labeled as above.

### 2.3. Preparation of Glass-Fiber-Reinforced APPEN Composite Laminates (APPEN/GF)

First, APPEN gel solution was fabricated for cloth dipping. Six layers of glass fiber were tailored and impregnated into the gel solution. When the fabrics were evenly impregnated, then the prepregs were placed in a blast oven and treated at 80 °C for 30 min to remove the solvent primarily. Prepregs were pretreated at 160 °C for 10 min to further homogenize the prepolymerization of APPEN. Finally, prepregs were added into the stainless steel mold with the pressure of 20 MPa at 180 °C for 1 h to obtain APPEN/GF composite laminates. The corresponding preparation process is presented in Figure 1. Owing to the different APPEN prepolymers, various APPEN/GF composite laminates are labeled as A5P4T20/GF, A6P4T40/GF, and A7P4T60/GF. Generally, the fiber volume fraction of the composite laminates was about 62%.

### 2.4. Characterizations

The curing behaviors of the prepolymers were investigated using differential scanning calorimetry (DSC) with a modulated DSC-Q100 (TA Instruments, New Castle, DE, USA). Heating rate was 10 °C/min and a constant nitrogen flow rate of 50 mL/min was set. The samples were tested from 50 °C to 300 °C. Gelation time (160, 180, and 200 °C) was tested by a gel time tester (YJ39-LA38-11BN, Shanghai, China) from Shanhai Yijia Electric Co., Ltd. Structures of APPEN prepolymers, prepregs, and composites were characterized using Fourier transform infrared spectroscopy (FT-IR, 8400S, Shimadzu, Kyoto, Japan). A SANS series microcomputer-controlled electronic universal testing machine (CMT6104, Shenzhen, China) was used to detected the mechanical properties of APPEN/GF composite laminates, and the mode used was three-point bending. Fracture surface morphology of APPEN/GF composites was detected with a scanning electron microscope (SEM, JSM25900LV, JEOL, Kyoto, Japan) operated at 20 kV. Dynamic mechanical analysis (DMA) was performed using DMA-Q800 (TA Instruments, USA) dynamic mechanical analyzer. The composite laminate samples were tailored to the dimensions of 40 mm * 10 mm * 1.5 mm and were heated from 50 °C to 300 °C at a heating rate of 5 °C/min. Thermogravimetric analysis (TGA) of the composite laminates was tested using TGA-Q50 (TA Instruments, New Castle, DE, USA). The samples were heated from 50 °C to 600 °C at a heating rate of 20 °C/min in a nitrogen atmosphere.

## 3. Results and Discussion

### 3.1. Polymerization Behaviors and Possible Polymerization Mechanisms of APPEN Prepolymers

In this work, a novel epoxy resin and an amino phthalonitrile co-polymerized for toughness with Poly (arylene ether nitrile) (PP-PEN) (shown in Figure 2) were fabricated. According to the literature [11,29,30], amine groups in APN molecular chains promote the ring-opening polymerization of benzoxazine rings and epoxy rings, which will generate generous hydroxyl groups. Then, through the ring-forming polymerization of phthalonitrile, copolymerization between the epoxy ring and amino segments and toughing aggregation structures is sequentially formed. In order to directly demonstrate the copolymerization processes of APPEN prepolymers, the gelation properties, DSC, and proposed polymerization mechanism of APPEN prepolymers were discussed.

#### 3.1.1. Gelation Properties of APPEN Prepolymers

The orthogonal calculation method was applied to reflect the gelation processes of APPEN prepolymers, shown in Table 2. According to the principle of orthogonal experiment, key accumulated values K_1_, K_2_, and K_3_ (K_b_ = ∑X_b_), the average values (k_b_ = K_b_/3) and the range (R = k_max_ − k_min_) of k values at each level were gained.

In our experiment, the content of epoxy resin, amino phthalonitrile, and prepolymerization time were concretely analyzed. For the factor A (content of epoxy resin), k_A2_ > k_A3_ > k_A1_ is presented, indicating that the level 1 of factor A was most conducive to the polymerization of the prepolymers. That is, by decreasing the content of epoxy resin, the polymerization rate can be elevated. Similarly, it can be determined that level 1 of factor B is the outstanding level, as well as level 3 of factor C. So, considering the prepolymerization conditions, the optimal combination of prepolymerization time and proportion of each component that affected the gelation time of APPEN prepolymer was identified as A5P4T20. The theoretical analysis revealed the primary and secondary order of each factor. As is known, with an increase in the range, the influence degree of the corresponding factors increases. The order of the range was R_B_ > R_C_ > R_A_, so the content of APN was determined as the most key factor that affects the gelation time of APPEN prepolymer, followed by the prepolymerization time.

To sum up, the content of APN and prepolymerization time demonstrated remarkable effects on the prepolymerization degree, while the effect of epoxy resin content was not obvious. At the same time, the effects of APN content (factor B) and prepolymerization time (factor C) on the gelation time of the APPEN prepolymers were k_B3_ > k_B2_ > k_B1_ and k_C1_ > k_C2_ > k_C3_, showing that increasing the content of APN and prepolymerization time were both conducive to promoting the reaction of APPEN prepolymers. However, considering the co-polymerization between epoxy rings and phthalonitrile without catalysis, the content of amino phthalonitrile was selected as a fourth factor. Then, to intuitively analyze the effects of phthalonitrile on the polymerization of epoxy resin, various prepolymers (A5P4T20, A6P4T40, and A7P4T60) were designed and the curing processes were discussed.

To further evaluate the polymerization processes and processing conditions, the gelation time of the prepolymers treated at 160 °C, 180 °C, and 200 °C was compared and analyzed, as shown in Figure 3. The gelation time of the prepolymers can be used as an indicator of polymerization rate. The shorter the gelation time, the faster the polymerization rate. Therefore, the results shown in Figure 3 indicated that testing temperature possessed significant influence on the gelation time of various prepolymers. With an increase in testing temperatures, the gelation time of the prepolymers sharply decreased. This can be attributed to the fact that catalytic polymerization between active amino groups and epoxy rings was promoted at higher temperatures [31]. With the increase in testing temperatures, the difference between the gelation times of various APPEN prepolymers reduced, indicating that processing temperatures possessed a significant influence on the polymerization, especially at high temperature ranges. However, with the increase in testing temperature, all of the APPEN prepolymers showed a short gelation time, which would be unfavorable to the subsequent molding processes [15]. Similarly, when the test temperature was set as 160 °C, the gelation time for all samples was too long (>300 s), which would result in the loss of the resin matrix during the molding processes. In sum, gelation tests were performed at 180 °C, which was determined as the corresponding molding process temperature.

#### 3.1.2. Curing Processes of APPEN Prepolymers

In this work, DSC was applied to analyze the polymerization processes of APPEN prepolymers because DSC curve shapes and enthalpy changes can reveal possible curing behaviors and polymerization mechanisms. Figure 4 shows the DSC curves of various prepolymers. It is obvious that two exothermic peaks appeared in all curves without obvious absorption peaks, which demonstrated that the various prepolymers possessed dual polymerization processes, corresponding to the ring opening of epoxy rings catalyzed by the amino groups and the ring forming of nitrile groups, respectively [20,32]. In a previous work [11], copolymerization between epoxy rings and amino phthalonitrile were investigated using dynamic rheology analysis. The results indicated similar double polymerization processes. With an increase in epoxy content and prepolymerization time extension, the exothermic peaks moved to the low temperature range, and the exothermic peak became less apparent, as shown in Table 3. With the increase in the content of epoxy resin and prolonging prepolymerization time, the peak temperature shifted from 202.04 °C to 190.87 °C, and the second peak gradually became apparent. It shifted from 252.11 °C to 257.00 °C, showing an opposite trend compared with the first peak in DSC curves. On the one hand, ring-opening polymerization of epoxy resin could be significantly triggered at about 200 °C, and with the increase in the content of epoxy resin, an obvious exothermic reaction appeared, resulting in conspicuous exothermic peaks. On the other hand, the ring-forming polymerization of nitrile groups occurred at high temperatures. The shift trend of exothermic peaks appearing at about 250 °C can be ascribed to the fact that the ring-forming polymerization of nitrile groups was suppressed, due to the copolymerization between amino groups and epoxy rings, which consumed a large amount of reactive hydrogen [5,11,29]. Furthermore, with the extension in prepolymerization time, the first exothermic peak became less steep, which can be attributed to the fact that the degree of copolymerization between epoxy resin and amino phthalonitrile was high during the fabrication of the prepolymers, resulting in the exothermic peak and exothermic enthalpy not being significantly detected using DSC.

#### 3.1.3. Possible Polymerization Mechanism of APPEN Prepolymers

Gelation time analysis and reaction process monitoring indicated that the amino and nitrile groups copolymerized with epoxy rings, and the ring-opening polymerization of epoxy rings was triggered by active amino structures. Therefore, the consumption of amino groups resulted in the sluggish ring-forming polymerization of nitrile groups. In our previous work, the copolymerization between epoxy and phthalonitrile containing benzoxazine was studied, and the results indicated that the introduction of epoxy significantly suppressed the ring opening of benzoxazine rings and ring-forming polymerization of phthalonitrile. Moreover, the active phenolic hydroxyl groups generated from the ring-opening reaction of oxazine rings promoted the polymerization of epoxy rings and formed copolymerized chain segment structures, which possessed outstanding thermal stability. To intuitively illustrate the characters of molecular structures, the FTIR spectra of the A5P4T20 prepolymer, A5P4T20 prepreg, and A5P4T20 composite are presented in Figure 5. The FTIR spectrum of APN was obtained in referred to our previous work [11]. It can be seen that the characteristic peaks of -NH_2_ (3342 cm^−1^) and -CN (2231 cm^−1^) appeared in all of the spectra but decreased in turn, indicating that along with the polymerization, corresponding groups were transformed. It was also obvious that characteristic peaks of triazine rings and phthalocyanine rings appeared at about 1488 cm^−1^ and 1005 cm^−1^, indicating the ring-forming polymerization of nitrile groups. Moreover, the peak of C-O-C appeared at 1085 cm^−1^, and for A5P7T20 composite, the peak became faint. This may be ascribed to the copolymerization between the epoxy ring and amino phthalonitrile, which would restrict the activity of the molecular and nitrile groups.

In sum, considering the DSC curves and gelation analysis, the possible polymerization processes of APPEN prepolymers can be summarized in Figure 6. Firstly, the amino groups in the main chain of APN trigger the ring opening of the epoxy ring (Figure 6a). Then, the active hydroxyl groups from the ring opening of epoxy rings and the residual active hydrogen derived from amino groups synergistically promote the ring-forming polymerization of nitrile groups (Figure 6b). The nitrile groups were mostly derived from amino phthalonitrile and the minuscule part of poly (arylene ether nitrile). Consecutively, APPEN-based polymers (Figure 6c) were obtained, which were composed of the backbone structure of epoxy matrix and aromatic heterocyclic structures, such as triazine rings, isoindole, and phthalocyanine rings, along with long chains of PP-PEN embedded in the network structures in the form of interpenetrating entanglement.

### 3.2. Mechanical Properties of APPEN/GF Composite Laminates

Mechanical properties of matrix-based composite laminates dominated the applications of the composites and were usually used to evaluate the molding processes and polymerization effects. Among them, flexural strength and modulus were the fundamental characteristics tested to evaluate the effects of components and prepolymerization time on the mechanical properties. As shown in Figure 7a, the flexural strength of the A5P4T20/GF, A6P4T40/GF, and A7P4T60/GF composites was 598, 712, and 496 MPa, respectively; meanwhile, flexural modulus was 28, 38, and 24 GPa, respectively, as shown in Figure 7b. All of the composite laminates possessed outstanding mechanical performances, and the A6P4T40/GF composite showed the optimal flexural strength and modulus. Moreover, when the content of epoxy resin and the prepolymerization time were increased, the flexural strength and modulus of the laminates first increased and then decreased. Generally, the resin matrix itself and the interfacial binding between the resin matrix and glass fibers mainly dominated the flexural strength and modulus of the laminates [20,31,33]. The higher the rigidity of the matrix, the better are the mechanical properties of the composite laminates, as well as the better is the interfacial interaction. For APPEN/GF composite laminates, the rigidity of the matrix was the key factor affecting the mechanical properties in this work. In the matrix system, the ring-opening polymerization of epoxy resin and ring forming of phthalonitrile contributed to an increase in polymerization degree and rigidity of the matrix. Therefore, the promotion of ring-opening and ring-forming polymerization dominated the mechanical performances of composite laminates. As described above, the ring-opening polymerization of the APPEN prepolymers was promoted by the appropriate ratio of epoxy resin and amino phthalonitrile. Moreover, the polymerization rate of the prepolymers could be adjusted by controlling the prepolymerization time. Subsequently, the degree of polymerization was improved. Generally, the higher the polymerization degree, the better is the mechanical performance. Thus, according to the results of the gelation and DSC tests, prolonging prepolymerization time would more obviously improve the polymerization degree to obtain the improved mechanical properties. However, by increasing the content of epoxy resin, the ring-opening polymerization of epoxy rings consumed many amino groups, which caused insufficient ring-forming polymerization on nitrile groups. Thus, A6P4T40/GF composite laminates possessed the most prominent flexural strength. A7P4T60/GF showed that decreasing mechanical properties can be ascribed to the relatively attenuated copolymerization degree between epoxy resin and amino phthalonitrile. Figure 7b shows the flexural modulus of the corresponding composite laminates. The results present a similar trend to that of flexural strength, indicating that the rigidity of the matrix and the interfacial compatibility between matrix and fibers also contribute to the flexural modulus of the laminates. In sum, improving the polymerization degree of the prepolymers and intensifying the interfacial binding between the matrix and fibers was conducive to the fabrication of high-mechanical-performance composite laminates.

To intuitively evaluate the interfacial binding between the matrix and fibers, the fracture morphology for the laminates was measured using SEM as shown in Figure 8. It was obvious that all of the fracture surface was rough and showed a ripple shape. This can be ascribed to the outstanding toughness of the APPEN matrix. According to the possible copolymerization mechanism, amino groups were crosslinked to the main chains of epoxy resin and phthalonitrile, and nitrile groups were embedded in the network in the form of aromatic heterocyclic rings, including triazine rings and phthalocyanine rings. Moreover, long chains of thermoplastic PEN run through the tree-dimensional structures, which would bring about the improvement in toughness. In Figure 8a,b, A5P4T20/GF composite laminates exhibited a relatively rough matrix surface and distinct ripple shape, demonstrating that the prepared A5P4T20 composites possessed high toughness [34]. Additionally, a great deal of resin was observed to adhere to the fiber surface, indicating that the energy absorption and stress transmission were concentrated mainly in the matrix and did not occur on the fiber surface [35]. Nevertheless, individual gaps can be observed in Figure 8(a1), which shows the stripping of matrix and fibers under stress load. In Figure 8b,c, ductile fractures and ripple shapes can also be detected, and the residual matrix is shown to adhere firmly to the fiber surface. No obvious gaps and hollows were presented in the vision field, manifesting the improvement of the interfacial compatibility. Nevertheless, compared with that of A6P4T40/GF, the fracture surface of A7P4T60/GF showed relative brittleness, due to the large and smooth matrix fracture section. Therefore, it can be concluded that the fracture surface of the composites can be significantly affected by an appropriate content and prepolymerization time, which would sequentially improve the brittleness and rigidity of the APPEN matrix and then achieve good interfacial compatibility.

### 3.3. Thermomechanical Properties of APPEN/GF Composites

To further evaluate the influence of copolymerization between epoxy and amino phthalonitrile resins on the thermomechanical properties of their laminates, the DMA test was employed. Generally speaking, the relationship between the storage modulus and the tangent delta (Tan δ) as a function of temperature would directly indicate the properties of the composites. In Figure 9, the APPEN/GF composite laminates with various contents of epoxy resin and pre-treated with various prepolymerization time exhibited great storage modulus (20~26 GPa), indicating that APPEN/GF composite laminates exhibit conspicuous rigidity. It is obvious that the storage modulus (26 GPa) of A6P4T40/GF laminate was faintly higher than that of A5P4T20/GF (24.7 GPa) and A7P4T60/GF (20 GPa), indicating that the appropriate epoxy content and prepolymerization time would dominate the storage modulus and rigidity of APPEN/GF laminates. This can be ascribed to the fact that the ring-opening polymerization of epoxy rings and ring-forming polymerization of nitrile groups were all catalyzed by amino groups. The polymerization of epoxy resin consumed many amino groups and restricted the polymerization of nitrile groups, although active hydrogen generated from the ring-opening polymerization of epoxy rings promoted the polymerization of nitrile groups in theory. Thus, appropriate content of epoxy resin and amino phthalonitrile were the key factors affecting the storage modulus and rigidity. Moreover, as mentioned above, the prepolymerization time of various APPEN prepolymers also showed obvious effects on the storage modulus of the composite laminates, which can be ascribed to the fact that copolymerization between epoxy rings and amino phthalonitrile was processed in stages. In the prepolymerization process, active amino groups triggered the ring opening of epoxy rings and amino phthalonitrile molecular chains were incorporated in the main chain of epoxy resin. The prepolymerization time dominated the crosslinking degree between the epoxy ring and amino phthalonitrile, which directly affected the length and configuration of the resulting molecular chains. Then, during the fabrication of composite laminates, crosslinking between amino and epoxy rings and interpenetrating entanglement of phthalonitrile and epoxy resin were further consolidated; that is, the length of molecular chains grew rapidly and the configurations of the aromatic heterocyclic structures were fastened quickly. Thus, the prepolymerization time directly determined the main structures of the composites. However, compared with that of the content of amino phthalonitrile, the effects of prepolymerization time on the storage modulus were weaker, due to the fact that the amino groups in phthalonitrile played an important role during the copolymerization.

Generally speaking, the peak temperature of tan *δ* plots was defined as *T_g_* of the composites. In Figure 9b, one conspicuous Tan δ transition peak and an indistinctive peak appeared, indicating that the APPEN/GF composite laminates may possess double glass transitions, as it was that weak phase separation that occurred during the fabrication of the composite laminates [36]. In addition, for A5P4T20/GF composites, the double *T_g_s* showed the highest values and the second Tan δ transition peaks were most obvious. The first *T_g_*_1_ increased from 175.85 °C (A7P4T60/GF) to 188.90 °C (A5P4T20/GF), and the *T_g_*_2_ possessed almost the same values (214.75 °C). This can be explained as follows: First, the ring-opening polymerization of epoxy rings in the system was promoted by amino groups and then the long backbone chains were formed. Due to the steric hinderance effects, the subsequent ring-forming polymerization of nitrile groups was restricted, showing a relatively low *T_g_*. Additionally, by increasing the content of amino phthalonitrile, the degree of copolymerization crosslinking was improved, and then higher *T_g_*s was gained. Additionally, it can be observed that by increasing the content of amino phthalonitrile, the peak intensity and peak width of tan δ plots decreased and became narrow. This may be ascribed to the fact that the motion of molecular chains was restricted due to the growth of molecular chains and corresponding steric hindrance, which would directly restrain the increase in polymerization degree. In summary, APPEN/GF composite laminates exhibited outstanding storage modulus and *T_g_,* which can facilitate the laminates playing an important role in high-performance mechanical material fields.

### 3.4. Thermal Stability of APPEN/GF Composite Laminates

The thermal stability of APPEN/GF composites was evaluated using TGA analysis, as shown in Figure 10, and the detailed thermal performance data (the temperature with thermal weight loss of 5% (*T*_5%_) and 10% (*T*_10%_), the residual carbon rate at 600 °C (*Y_c_*), and the integral program decomposition temperature (*IPDT*)) are shown in Table 4. The *IPDT* was composed of Formula (1) and is shown in Figure 11 [37,38,39,40]:(1)$$IPDT=AK\times (T_{f}-T_{i})+T_{i}$$

*A* and *K* are the area ratios defined by the TGA analysis curve shown in Figure 11, *T_f_* is the final experimental temperature (590 °C), and *T_i_* is the initial experimental temperature (50 °C). The *A* and *K* were calculated by Formulas (2) and (3) [38,39,40,41].
(2)$$A=\frac{S_{1}+S_{2}}{S_{1}+S_{2}+S_{3}}$$
(3)$$K=\frac{S_{1}+S_{2}}{S_{1}}$$

From Figure 10 and Table 4, it can be seen that the *T_5%_* of almost all APPEN/GF composites was higher than 334 °C, *T*_10%_ for all was higher than 360 °C, all *Y_c_* were up to 50% at 600 °C, and *IPDT* was higher than 1482 °C, indicating that the cured products possessed outstanding thermal stability. By adjusting the content of epoxy resin and prolonging the prepolymerization time, *T*_5%_ of A5P4T20/GF composites showed the highest value (348.62 °C), while *Y_c_* of A7P4T60/GF composites presented the maximum (58.28%). This can be ascribed to the step polymerization between epoxy resin and amino phthalonitrile. As described above, the ring-opening polymerization of epoxy rings was preferentially catalyzed by amino groups and elementary copolymers were formed. Then, long chain stacking and the three-dimensional network structures of the composites were fabricated. During the polymerization processes, the relative content of epoxy resin and amino phthalonitrile would dominate the main molecular chain of the copolymers, which affected the initial decomposition temperature (*T*_5%_), and the prepolymerization time directly influenced the aggregation configuration of the main chains and aromatic heterocyclic structures, which dominated the char yield of the composites (*Y_c_*). Moreover, all of the composite possessed high *IPDT* values, which also indicated that the epoxy resin and amino phthalonitrile matrix were well combined, and there was no volatilization or degradation at high temperatures, which would result in reduced thermal stability. In sum, the high *IPDT* and outstanding thermal stability once again proved the good copolymerization between epoxy resin and amino phthalonitrile, which generated a stable aromatic heterocyclic crosslinking network [38].

## 4. Conclusions

In this work, copolymerization between epoxy resin and amino phthalonitrile matrix was designed, and the effect of relative contents and molding processes on the pre-polymerization and polymerization reaction behaviors were systematically analyzed. The results indicated that prepolymerization time and component content played different roles during the polymerization processes, and the component content of amino phthalonitrile exhibited the most significant influence on the polymerization of APPEN composites. The structures of cured APPEN composites mainly consisted of a phenolic epoxy backbone chain, isoindole, triazine ring, and phthalocyanine ring. Ascribed to the improved crosslinking density and polymerization degree of APPEN prepolymers, their glass-fiber-reinforced laminates presented enhanced flexural strength (712 MPa) and modulus (38 GPa). The investigation of fracture morphology indicated that APPEN/GF composites were toughened efficiently, and the interfacial binding between matrix and fiber was intensified by controlling the matrix constituent and introducing the thermoplastic PP-PEN. Additionally, A5P6T20/GF composites possessed outstanding thermomechanical properties and excellent thermal stability (*T_g_* > 185 °C, *T*_5%_ > 334.89 °C, *Y*_c, 600 °C_ > 50%, and *IPDT* > 1482 °C). In summary, convenient and effective modification methods for the improvement of epoxy matrix high-performance composites and their laminates were provided.

## Figures and Tables

**Figure 1 polymers-15-03516-f001:**
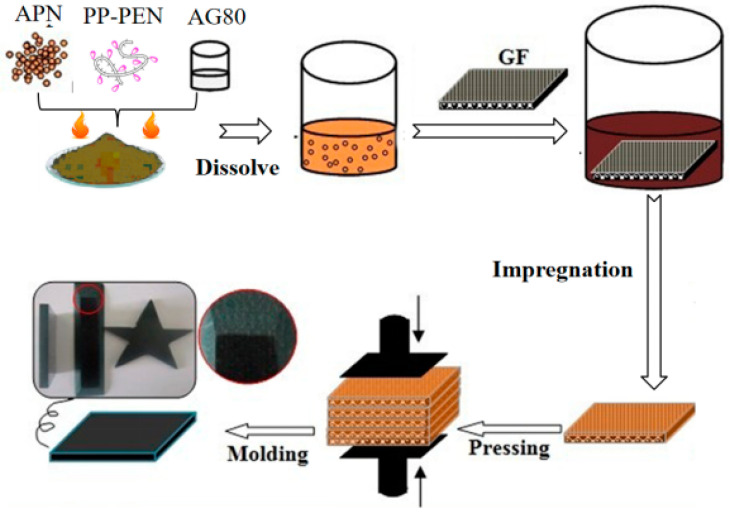
Preparation process of the fiber-reinforced composite laminate.

**Figure 2 polymers-15-03516-f002:**
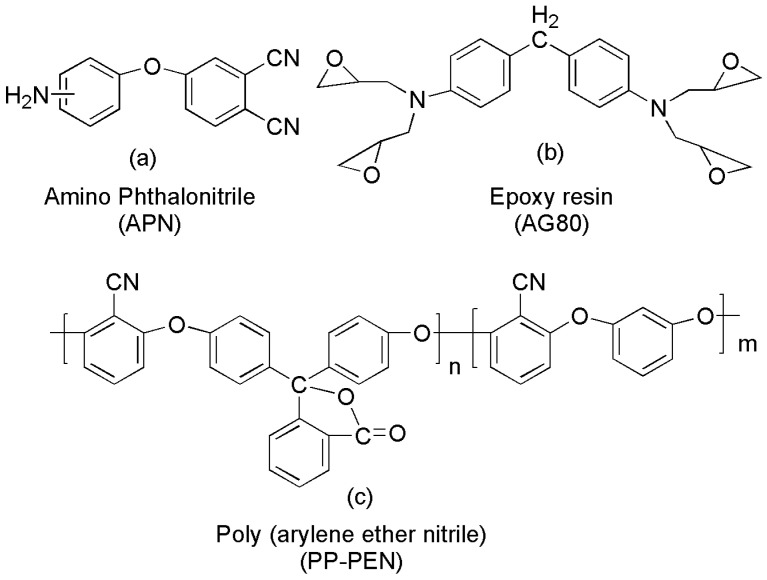
Chemical structure of (**a**) amino phthalonitrile, (**b**) epoxy resin, and (**c**) Poly (arylene ether nitrile).

**Figure 3 polymers-15-03516-f003:**
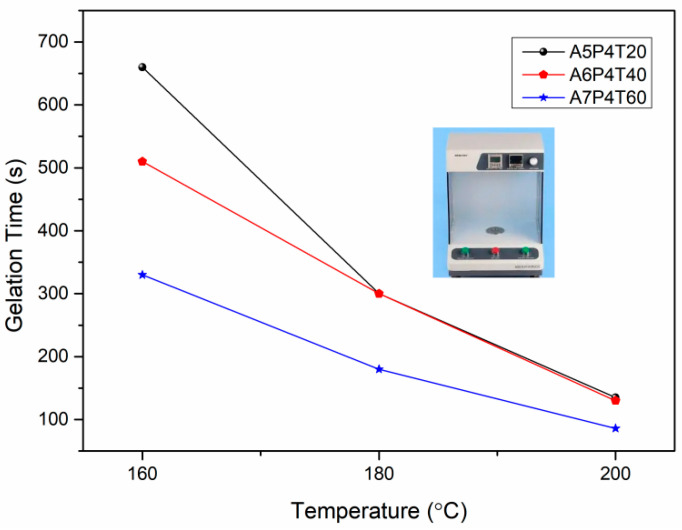
Gelation tests of APPEN prepolymers at various temperatures.

**Figure 4 polymers-15-03516-f004:**
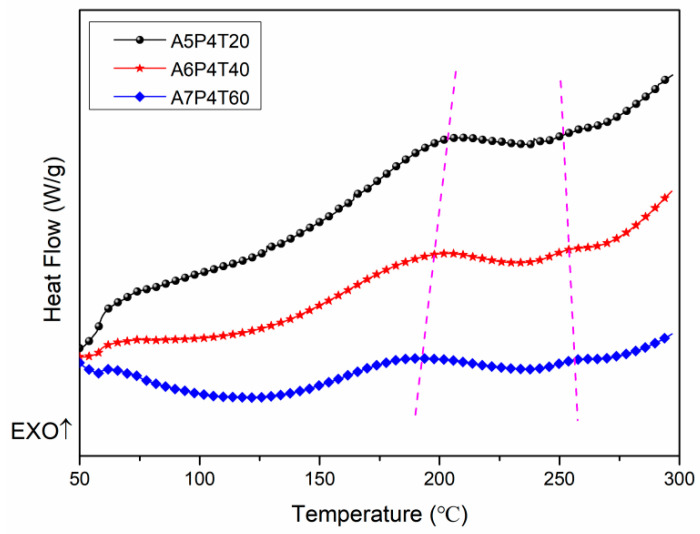
The DSC analysis curves of APPEN prepolymers.

**Figure 5 polymers-15-03516-f005:**
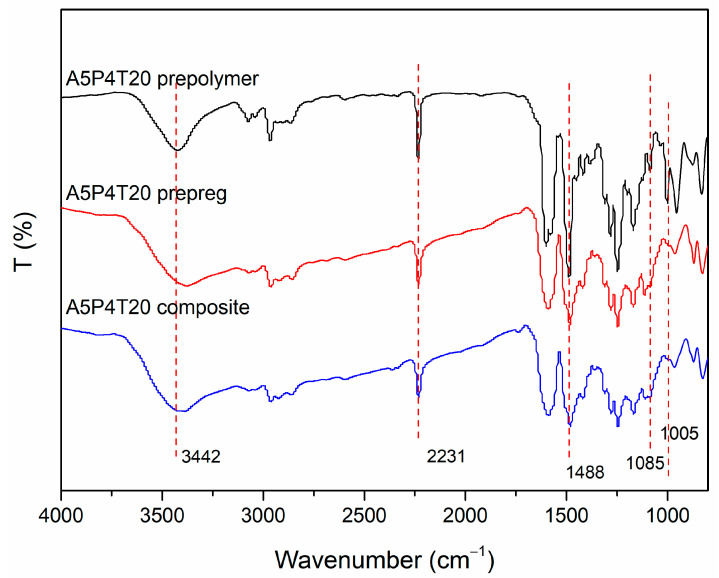
FTIR spectra of APPEN resin at various polymerization stages.

**Figure 6 polymers-15-03516-f006:**
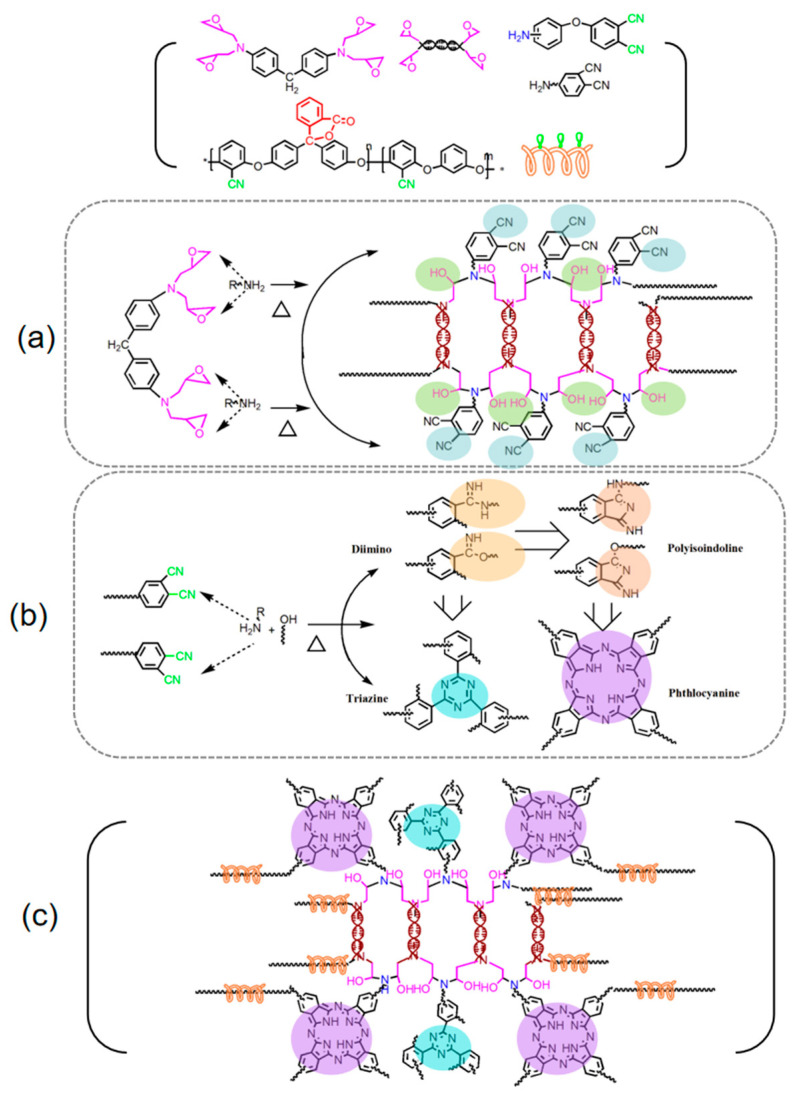
Possible curing reaction mechanism of APPEN prepolymers: (**a**) ring-opening polymerization of epoxy rings catalyzed by amino groups, (**b**) ring-forming polymerization of nitrile groups catalyzed by the active hydroxyl and amino groups, and (**c**) possible network structures of APPEN composites.

**Figure 7 polymers-15-03516-f007:**
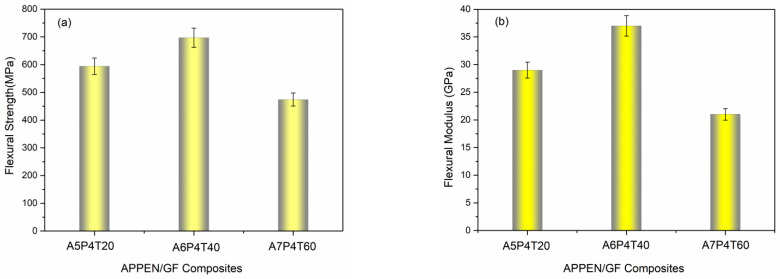
(**a**) Flexural strength and (**b**) flexural modulus of APPEN/GF composite laminates.

**Figure 8 polymers-15-03516-f008:**
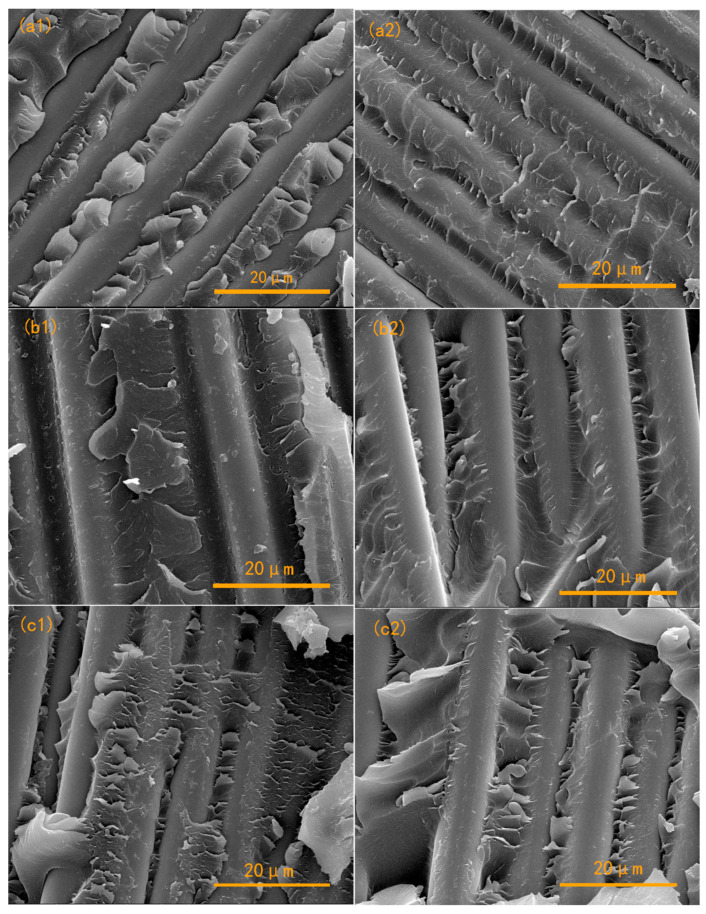
Fracture morphology of APPEN/GF composite laminates: (**a**) A5P5T20/GF, (**b**) A6P4T40/GF, and (**c**) A7P4T60/GF.

**Figure 9 polymers-15-03516-f009:**
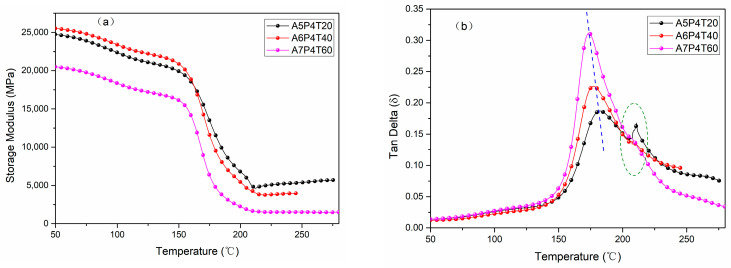
Thermomechanical analysis curves of APPEN/GF composites: (**a**) Storage modulus and (**b**) Tan delta.

**Figure 10 polymers-15-03516-f010:**
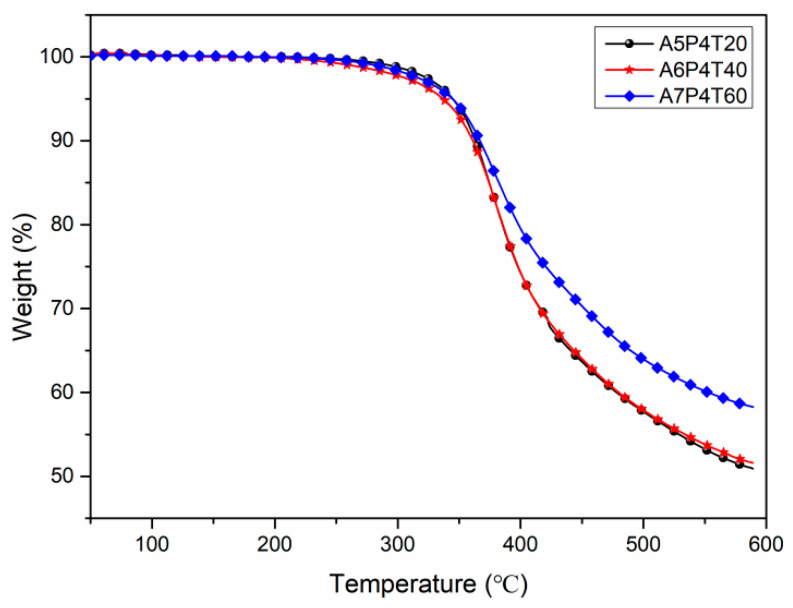
Thermal stability of APPEN/GF composites.

**Figure 11 polymers-15-03516-f011:**
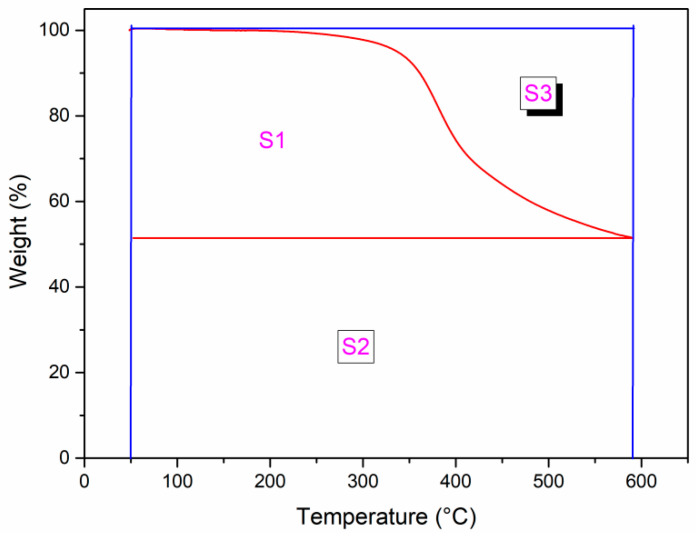
Diagram of S_1_, S_2_, and S_3_ in the Formula (1). Red lines represent the corresponding TGA curves and horizontal line of char yield; blue lines indicate the horizontal line of initial weight and the vertical line of the initial and final measured temperatures.

**Table 1 polymers-15-03516-t001:** Table of the various prepolymers including content and prepolymerization time.

Samples	A Weight of AG80 (g)	B Weight of APN (g)	C Prepolymerization Time (min)
1	5	4	20
2	5	3	40
3	5	2	60
4	6	4	40
5	6	3	60
6	6	2	20
7	7	4	60
8	7	3	20
9	7	2	40

**Table 2 polymers-15-03516-t002:** Orthogonal experiment of APPEN prepolymers (at 180 °C).

Samples	A Weight of AG80 (g)	B Weight of APN (g)	C Prepolymerization Time (min)	Gelation Time (s)
1	5	4	20	300
2	5	3	40	300
3	5	2	60	300
4	6	4	40	300
5	6	3	60	360
6	6	2	20	600
7	7	4	60	180
8	7	3	20	480
9	7	2	40	510
K1	900	780	1380	R_B_ > R_C_ > R_A_
K2	1260	1140	1110	
K3	1170	1410	840	
k1	300	260	460	
k2	420	380	370	
k3	390	470	280	
	k2 > k3 > k1	k3 > k2 > k1	k1 > k2 > k3	
R	120	210	180	

**Table 3 polymers-15-03516-t003:** The thermal properties of APPEN prepolymers derived from DSC curves.

Samples	*T_top_*_1_ (°C)	*T_top_*_2_ (°C)
A5P4T20	202.04	252.11
A6P4T40	198.65	253.71
A7P4T60	190.87	257.00

**Table 4 polymers-15-03516-t004:** Detailed thermal performance data of APPEN/GF composite laminates.

Sample	*T* _5%_	*T* _10%_	*Y_c_*	*IPDT*		
	(°C)	(°C)	(%, 600 °C)	*A*	*K*	*T* (°C)
A5P4T20	348.62	363.56	50.94	0.8390	2.538	1199.98
A6P4T40	334.89	360.08	51.89	0.8380	2.603	1227.90
A7P4T60	346.67	370.14	58.28	0.8638	3.070	1482.18

## Data Availability

The raw/processed data required to reproduce these findings cannot be shared at this time as the data also form a part of an ongoing study.

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
