# Peer review of "Study of the Self-Polymerization of Epoxy/Phthalonitrile Copolymers and Their High-Performance Fiber-Reinforced Laminates"

_polymers, 2023, doi:10.3390/polym15173516_

Round 1

Reviewer 1 Report

The paper provides useful information regarding self-polymerization of a kind of copolymer and then using it in fiber laminated composites. The results are well-presented. I have some minor concerns that need to be addressed.

1- How much was the fiber volume fraction and how did you measure that?

2- It would be beneficial if authors can schematically demonstrate the process of sample preparation.

3- I strongly recommend the authors to provide FTIR in the results section.

4- I am quite surprised that why authors did not measure other mechanical properties like tensile strength, Young's modulus, Interfacial shear strength. Could you please add them to the manuscript?

N/A

Author Response

Response to Reviewer 1 Comments

Point 1:  How much was the fiber volume fraction and how did you measure that?ρρρ

Response 1: In this manuscript, the relative content of fibers and the resin matrix was directly expressed by weight percent. For the prepared composite laminates, the fiber volume fraction can be obtained by hydrometer algorithm. According to the corresponding density of fibers (f=2.63g/cm3) , the resin matrix (=1.050g/cm3) and the composites (=2.280g/cm3), the fiber volume fraction of the composite laminates was about 65.2%, referring to the following calculation formula.

. Fiber volume fraction =

Point 2:  It would be beneficial if authors can schematically demonstrate the process of sample preparation

Response 2: Thanks for your advice. The preparation process of the composite laminate was replenished in the revised manuscript (2.3. Preparation of Glass Fiber-Reinforced APPEN Composite Laminates (APPEN/GF )) and copied here.

Figure 1 Preparation process of the fiber-reinforced composite laminate

Point 3: I strongly recommend the authors to provide FTIR in the results section.

Response 3: Thanks very much for your suggestion. For polymer composites, the polymerization processes were accompanied by structure changes, which can be monitored by FTIR. Thus, to intuitively illustrate the characters of molecular structures, the FTIR spectra of A5P4T20 prepolymer, A5P4T20 prepreg and A5P4T20 composite were replenished in the revised manuscript. And the corresponding discussion was copied here.

“……To intuitively illustrate the characters of molecular structures, the FTIR spectra of A5P4T20 prepolymer, A5P4T20 prepreg and A5P4T20 composite were presented in Figure 4. FTIR spectrum of APN was referred to our previous work [11]. It can be seen that the characteristic peaks of -NH2 (3342 cm-1) and -CN (2231 cm-1) appeared in all of the spectra but decreased in turn, indicating that along with the polymerization, corresponding groups were transformed. Also, it’s obvious that characteristic peaks of triazine rings and phthalocyanine rings appeared at about 1488 cm-1 and 1005 cm-1, indicating the ring-rorming polymerization of nitrile groups. Moreover, the peak of C-O-C appeared at 1085 cm-1 and for A5P7T20 composite, the peak became faint. It may be ascribed to the copolymerization between epoxy ring and amino phthalonitrile, which would restrict the activity of the molecular and groups.”

Figure 4. FTIR Spectra of APPEN resin at various polymerization stages

Point 4: I am quite surprised that why authors did not measure other mechanical properties like tensile strength, Young's modulus, Interfacial shear strength. Could you please add them to the manuscript?

Response 4: We wholeheartedly agree with that more measurements for mechanical properties including tensile strength, Young’s modulus, interfacial shear strength, impact properties and so on, would be in favor of the investigation on the copolymerization processes and crosslinking network structures of APPEN/GF composites. To assist the results of machaical properties, dynamic thermal mechanical properties were investigated in the manuscript (revised as Figure 6). In our preparing work, the detail investigation for the relationship and influence between the mechanical properties and the crosslinking network structures were performed, and the research results of interlaminates shear strength and impact properties for APPEN composites were in preparing. As your suggestion,  the measurements of tensile strength and Young’s modulus would also be carried out and will be presented in the following research work.

Author Response

Response to Reviewer 2 Comments

Point 1: Author needs to revise full manuscript as I can see error like similar words repetition (line 164) and various grammatical errors.

Response 1: Thanks for your advise. In this manuscript, grammatical errors would affects readers' understanding and evaluation. We have seriously checked refined the manuscript thoroughly. Also, several colleagues who are skilled authors of English language papers were asked to revise the manuscript.

Point 2: Need to include some of the latest study in the Introduction and in result and

appropriate references to discrepancies and facts in result and discussion section.

Response 2: Thank you for your advice on the references in the manuscript. References 6,7,8,9 and 10 have been replaced by new references appeared to be the latest references for the investigation of epoxy resin-based composites and main modification methods. Additional, relative references have been replenished in the discussion section to further illustrate the discrepancies and facts. And some revisions were copied here.

“……It’s obvious that two exothermic peaks were appeared in all curves without obvious absorption peaks, which demonstrated that the various prepolymers possessed dual polymerization processes, corresponding to the ring-opening of epoxy rings catalyzed by the amino groups and ring-forming of nitrile groups, respectively [33, 34]. In the previous work [11], copolymerization between epoxy rings and amino phthalonitrile were investigated by dynamic rheology analysis. Results indicated the similar double polymerization processes……”

“……Additionally, abound resin was observed to adhere to the fiber surface, indicating that the energy absorption and stress transmission were concentrated mainly in the matrix and did not occur on the fiber surface [37]……”

Reference:

6. Seo, H.Y.; Im, D.; Kwon Y.J.;  Nam, C.Y.; Kim, S. H.; Nam, T.; Kim, C.;  Vivek, E.; Baek, K.Y.; Cho, K.Y.; Yoon, H.G. A strategy for dual-networked epoxy composite systems toward high cross-linking density and solid interfacial adhesion. Compos. Part B 2023, 254, 110524

  1. Wang, Z.L.;Hou, D.J.; Wang, F.; Zhou, J.J.; Cai, N.; Guo, J. Facile and Scalable Strategy for Fabricating Highly Thermally Conductive Epoxy Composites Utilizing 3D Graphitic Carbon Nitride Nanosheet Skeleton. ACS Appl. Mater. Interfaces 2023, 15, 23, 28626–28635
  2. Ayyanar, C.B.;Marimuthu, K.; Mugilan, T.; Gayathri, B.; Sanjay, M.R.; Khan, A.; Siengchin, S. Novel Polyalthia Longifolia seed fillers loaded and E-glass fiber-reinforced sandwich epoxy composites. P I Mech Eng E-J Pro 2023, 0,0
  3. Zhang, R.;Wang, H.S.; Wang, X.Z.; Guan, J.; Li, M.Q.;  Chen, Y.F. Rubber-Composite-Nanoparticle-Modified Epoxy Powder Coatings with Low Curing Temperature and High Toughness. Polymers 2023, 15,195
  4. Taieh, N.K.; Khudhur, S.K.; Fahad, E.A.; Zhou, Z.W.; Hui, D.V. High mechanical performance of 3-aminopropyl triethoxy silane/epoxy cured in a sandwich construction of 3D carbon felts foam and woven basalt fibers. Nanotechnology Reviews2023, 12, 519

Point 3:  I would like to understand why author selected 180 C instead of 160 C when we know increasing the temperature decreases the gelation time and that could be unfavorable.

Response 3: What you concerned was elaborately discussed in our research group. In this work, the copolymerization between epoxy resin and amino phthalonitrile was investigated with gelation time testing. Results indicted that processing temperatures possessed significantly influence on the polymerization, especially at high temperature ranges. However, with increasing the testing temperature, all of APPEN prepolymers showed short gelation time, which would be unfavorable to the subsequent molding processes. Similarly, when the test temperature was set as 160 ℃, gelation time for all samples was too long (>300 s), which would result in the loss of resin matrix during the moulding processes. In sum, gelation tests were performed at 180 ℃, which was determined as the corresponding molding process temperature.

Point 4: Can author provide further details and additional supporting details on individual gaps appeared in Figure 6 (a1) and ductile fractures and ripple shapes in Figure 6 (b) and (c)?

Response 4: Thanks for your reminding. Figure 6 (revised as Figure 8) showed the fracture appearance of various composite laminates. In the discussion section, the individual gaps were ascribed to the stripping of matrix and fibers under stress load. And the ductile fractures appearances and ripple shapes were also detected, which were attributed to the improvement of interfacial compatibility. To assist the predicted results of ductile fractures, in our preparing work,  detail investigation of the interlaminates shear strength and impact properties for the APPEN composites have been carried out and will be presented in the following research work.
